# Cardiac and Locomotor Responses to Acute Stress in Signal Crayfish *Pacifastacus leniusculus* Exposed to Methamphetamine at an Environmentally Relevant Concentration

**DOI:** 10.3390/ijerph17062084

**Published:** 2020-03-21

**Authors:** Filip Ložek, Iryna Kuklina, Kateřina Grabicová, Jan Kubec, Miloš Buřič, Tomáš Randák, Petr Císař, Pavel Kozák

**Affiliations:** Faculty of Fisheries and Protection of Waters, South Bohemian Research Center of Aquaculture and Biodiversity of Hydrocenoses, Zátiší 728/II, University of South Bohemia in České Budějovice, 389 25 Vodňany, Czech Republic; ikuklina@frov.jcu.cz (I.K.); grabicova@frov.jcu.cz (K.G.); kubecj@frov.jcu.cz (J.K.); buric@frov.jcu.cz (M.B.); trandak@frov.jcu.cz (T.R.); cisar@frov.jcu.cz (P.C.); kozak@frov.jcu.cz (P.K.)

**Keywords:** aquatic environment, invertebrates, illicit drug, behaviour, predator–prey relationship, pollution

## Abstract

Methamphetamine (METH), a central nervous system stimulant used as a recreational drug, is frequently found in surface waters at potentially harmful concentrations. To determine effects of long-term exposure to environmentally relevant levels on nontarget organisms, we analysed cardiac and locomotor responses of signal crayfish *Pacifastacus leniusculus* to acute stress during a 21-day exposure to METH at 1 μg L^−1^ followed by 14 days depuration. Heart rate and locomotion were recorded over a period of 30 min before and 30 min after exposure to haemolymph of an injured conspecific four times during METH exposure and four times during the depuration phase. Methamphetamine-exposed crayfish showed a weaker cardiac response to stress than was observed in controls during both exposure and depuration phases. Similarly, methamphetamine-exposed crayfish, during METH exposure, showed lower locomotor reaction poststressor application in contrast to controls. Results indicate biological alterations in crayfish exposed to METH at low concentration level, potentially resulting in a shift in interactions among organisms in natural environment.

## 1. Introduction

Aquatic environments are contaminated by a wide variety of pharmaceutically active compounds (PhACs) including illicit drugs such as opium, peyote, coca leaf and its derivatives heroin and cocaine, as well as synthetic amphetamine-type stimulants [1] usually used as recreational drugs [2]. Amphetamine-type stimulants are the third most commonly used recreational illicit drug after cannabis and opioids worldwide [3]. Methamphetamine (METH) has historically been popular in Central Europe due to its low cost, and has more recently appeared in Cyprus, Germany, Spain, Northern Europe and, especially, in Asia and North America [3,4,5].

Similar to other pharmaceutically active compounds, residues of illicit drugs originating from human excretion, due to their chemical and physical properties, may leave wastewater treatment plants [6] unaltered, or slightly transformed, before reaching surface waters [7,8]. Concentrations of these drug residues are reported to reach levels from ng to μg L^−1^ in wastewater and surface waters [9,10,11,12,13,14,15]. At these concentrations, illicit drug residues are not considered likely to be acutely toxic to nontarget organisms, i.e., organisms which are exposed unintentionally.

Drug residues may be integrated into the tissue of aquatic invertebrates [16] and vertebrates [17], and can modify the physiology of invertebrates, i.e., zebra mussels [18,19,20]. Planarians [21,22] and crayfish [23,24,25,26,27] showed, similarly like target organism, reinforcing effect to higher concentrations of dopaminergic drugs. Impact of drugs to physiology and behaviour were reported by Liao et al. (2015) [28] on aquatic vertebrates such as fish *Oryzias latipes* and European eel *Anguilla anguilla* [29].

Effects of METH on mammals are well documented [30,31], whereas information of its impact on nontarget aquatic invertebrates at environmentally relevant concentrations is limited. Exceptions are Hossain et al. [32] who observed disrupted sheltering behaviour in METH-exposed crayfish and Guo et al. [33] who reported disrupted burrow excavation in METH-exposed crayfish. The crayfish is a promising model in investigating physiological and behavioural alterations associated with low concentrations of PhACs [34,35,36]. In this study, our aims were to quantify cardiac and locomotor responses to acute stress in signal crayfish *Pacifastacus leniusculus* during, and following, exposure to environmentally relevant concentrations of METH and to infer the potential impact of the drug residues on the wider aquatic ecosystem, particularly with respect to predator–prey relationships.

## 2. Material and Methods

### 2.1. Experimental Animals, Experimental Setup, Data Acquisition, Chemicals

Signal crayfish *P. leniusculus* were collected in spring 2018 from Křesanovský Brook (49°03′35.2″ N 13°45′33.8″ E) positioned outside the possible sources of pollution, flowing from Šumava National Park, Czech Republic. Crayfish were transferred to a recirculation system at the Faculty of Fisheries and Protection of Waters USB, Czech Republic. We separated 12 males and 12 females (carapace length 44.8 ± 0.82 mm) in intermoult, and divided into a METH exposure group and a METH-free control group (50:50 sex ratio). A noninvasive sensor for monitoring cardiac activity was attached to each crayfish as described by Kuklina et al. [37], and then the fish were placed individually into 10 L static tanks containing shelters made from half of a plastic plant pot with a narrow oblong opening cut in the upper part to ensure unimpeded entrance of the crayfish carrying sensors. Opaque walls of tanks prevented disturbance; the tanks were maintained at constant water temperature of 20 °C and a 12:12 dark: light regime, arranged in an arena system with a single video camera. Crayfish were fed on chironomid larvae every 48 h. 

Cardiac activity, as heart rate (HR), and locomotor activity, as distance moved (cm), time spent in locomotion (%), and velocity (cm s^−1^) in response to a natural stress odour were measured on Days 0, 1, 7, 14, and 21 of drug exposure and Days 1, 2, 7, and 14 of a depuration phase in both METH-exposed and METH-free control groups. Heart rate and locomotor responses were recorded for the 30 min before and 30 min after stress application. The stress stimulus, scent of an injured conspecific to simulated predation, consisted of haemolymph drawn from an anaesthetised crayfish cut sagittally in 2 L dechlorinated water. Haemolymph was distributed at 50 mL per tank by a system of peristaltic pumps over the course of 1 min [38]. Video recordings were processed using EthoVison XT software (Noldus Information Technology, Wageningen, The Netherlands).

A stock solution of METH (Sigma Aldrich, Darmstadt, Germany) was prepared at 10 mg L^−1^ in ultrapure water, and stored at 4 °C. The METH exposure bath was prepared by adding stock solution to aged tap water to obtain a concentration of 1 µg L^−1^. The exposure solution was renewed every 48 h. Four times during exposure and four times during depuration, at time 0 after METH solution addition and at time 48 h before the water change, water was sampled by plastic syringe, filtered through 0.20 µm regenerated cellulose (Labicom, Prague, CR), and stored at −20 °C until analysis. The concentration of METH in water was determined by liquid chromatography with tandem mass spectrometry (TSQ Quantiva, Thermo Fisher Scientific, San Jose, CA, USA) using isotope dilution with D_5_-METH from Cerilliant as isotopically labelled internal standard as described by Hossain et al. [32].

### 2.2. Statistical Analysis

Data were evaluated using Statistica 13 (StatSoft Inc., Tulsa, OK, USA). In cases of non-normality of data distribution and heterogeneity of variance, nonparametric tests were used. Mean crayfish HR pre- and poststimulus were compared using a paired *t* test for dependent samples with results expressed as percentage difference. Differences in HR of control and exposed crayfish were evaluated by *t* test for independent samples. Distance moved, velocity, and time spent in locomotion were compared by nonparametric Mann–Whitney and Kruskal–Wallis tests before and after stress exposure as well as between control and METH-treated groups during exposure and depuration. Results were considered significant when *p* < 0.05.

## 3. Results

### 3.1. Analysis of Water Samples

The concentrations of METH in analysed water samples (n = 4) was 1.5 ± 0.1 μg L^−1^ at time 0 (freshly prepared solution) and 1.3 ± 0.4 μg L^−1^ after 48 h. Water of the control group showed METH concentration lower than the limit of quantification (< 0.04 μg L^−1^).

### 3.2. Cardiac Activity

During the exposure and depuration periods, mean HR of METH-exposed crayfish demonstrated weaker poststress cardiac reaction, except on Day 21 of the exposure when HR of groups did not differ (*p* > 0.05) and Day 2 of depuration when the mean HR poststress of the exposed group was significantly higher than that of controls (Figure 1).

#### 3.2.1. Exposure Period

In 61% of recordings, the mean HR of individual specimens during the exposure period significantly increased (*p* < 0.05) after stressor application in METH-exposed (Figure 2A) compared to 73% in the control crayfish (Figure 2C). Following stress application, 33% of METH-exposed crayfish and 25% of control crayfish demonstrated significantly decreased HR, while 6% of exposed and 2% of controls showed no significant cardiac response to stress odour.

#### 3.2.2. Depuration Period

Significant increase (*p* < 0.05) of HR after stressor application was detected in 56% of recordings in previously METH-exposed crayfish (Figure 2B) compared to 83% in the control group (Figure 2D) during depuration period. In 29% of previously exposed and 13% of controls, crayfish demonstrated a significant decrease in HR after stress application. No significant cardiac response to stress stimulus was detected in 15% of recordings of previously exposed crayfish and in 4% of control recordings. A single moulted crayfish in the control group was not included in analysis of the two final sampling days.

Information of individual mean HR of exposed and control crayfish over 30 min pre- and poststress during exposure and depuration periods is shown in the Appendix A).

### 3.3. Locomotion

Locomotor activity of METH-exposed crayfish did not significantly differ from that of controls (*p* > 0.05). However, crayfish within control group showed significant (*p* < 0.05) increase of distance moved (Figure 3) and velocity (Figure 4) in response to stress during exposure period in contrast to the METH-exposed group where distance and velocity before and after stress application did not differ.

## 4. Discussion

Despite Hazlett (1994) [39] found that the crayfish *Orconectes propinquus* did not show an alarm response to crushed conspecifics, while *O. virilis* could show strong feeding response, or a mixture of alarm and feeding responses to crushed crayfish, depended on how the crayfish was prepared. We used fluids of injured conspecifics to provide a scent associated with predation based on our previous experiences that has been shown to stimulate locomotion and cardiac activity in greater degree than other natural stressors, i.e., food [35]. So, the stressor effect was confirmed for signal crayfish as well as for method of preparation of the stressor cue. 

Cardiac activity expressed as percentage difference in mean heart rate pre- and poststimulus in METH-exposed group showed significantly lower stress reaction compared to controls (Figure 1), in contrast to the reported effect of the synthetic opioid tramadol on crayfish cardiac activity [38].

Despite altered physiological patterns, the locomotor activity of METH-exposed crayfish did not differ from that of the controls, which could be explained by high variation of individual locomotor reaction to stimuli (Figure 3 and Figure 4). Hossain et al. [32] exposed marbled crayfish to the same concentration of methamphetamine as used in the present study, and reported similar results with respect to locomotion, but found reduced sheltering behaviour in the exposed group. However, we found the reaction to a natural stressor to be lower compared to that observed in unexposed crayfish, implying increased risk of predation, especially when combined with lower shelter use [32]. This may lead to disruption of predator–prey balance and indicate potential for food web and biodiversity alteration, as was observed by Bláha et al. [40] in a study of predatory insects. Guo et al. [33] reported female METH-exposed crayfish excavated burrows of lower depth and volume relative to individual weight than did controls, which had possible negative consequences especially during periods of drought. In general, disturbance in behaviour or a physiological process at individual-to-population level leads to the disruption of related functions/system and breakdown of ecosystem processes [41].

Planarians have been used as a simple neural system model to study reinforcement effects of licit and illicit drugs [21,22], and crayfish as a model organism to study sensitivity of their reward system [26]. Our findings were in contrast to those of Imeh-Nathaniel et al. [24,42], who observed METH injected into crayfish to increase locomotor activity. In these studies, a higher concentration of METH was used and was injected directly into neural tissue. Changes induced by METH are likely dose specific.

Alteration of cardiac activity during acute stress, and its previously reported neurotoxicity [43] and cardiotoxicity [44], suggests a possible link to METH disruption of catecholamine production/reception in aquatic organisms. Moreover, under natural conditions, organisms are often exposed to a cocktail of contaminants [16]. Ascertaining the impact of such mixtures on nontarget organisms is complex, requiring prediction of synergistic or antagonistic mechanisms of action of individual components. 

Organism sensitivity to chemical compounds and natural stimuli (phenotypic plasticity) is considered an evolutionary mechanism to deal with dynamic changes in environmental conditions. In addition, species- and dose-specific effects, as reported by Brodin et al. [41] and Buřič et al. [45], of particular PhACs further complicate the situation. Many substances persist, and may bioaccumulate, in freshwater ecosystems that have diverse effects on organisms, communities, and entire ecosystems [16,41], potentially leading to substantial biodiversity loss and habitat changes. 

## 5. Conclusions

Impact on crayfish biology of METH at environmentally relevant concentrations is apparent in the recorded cardiac and locomotor responses to acute stress and could influence predator–prey relationship with potential ecological consequences. Alteration in physiological processes lead to the disruption of related functions of an organism, which may modify larger scale ecosystem processes [41]. Further research is needed to provide information on the risks of micropollutants to freshwater ecosystems.

## Figures and Tables

**Figure 1 ijerph-17-02084-f001:**
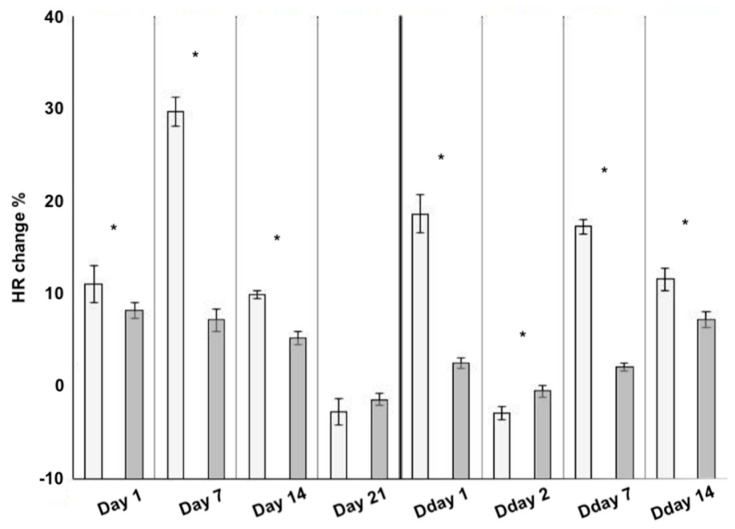
Mean heart rate change in control (light grey) and methamphetamine-exposed (dark grey) signal crayfish (n = 12) pre- and poststress initiation at each sampling day (Day 1–Day 21 days of exposure and Day 1–Day 14 days of depuration). * indicates significant difference (*p* < 0.05). Columns correspond to the mean value and whiskers correspond to standard error of mean.

**Figure 2 ijerph-17-02084-f002:**
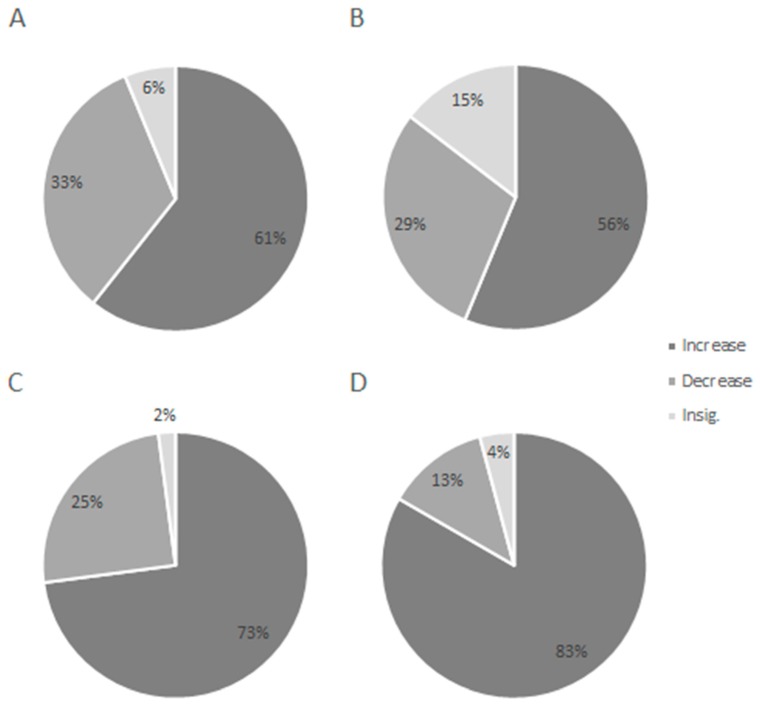
Cardiac response, as percent of recordings showing a significant change in mean heart rate of signal crayfish over 30 min after exposure to stress stimulus compared to the 30 min before stimulus. **A** = METH exposure, **B** = METH exposure depuration, **C** = control exposure, and **D** = control depuration. Four trials during exposure and four during the depuration period × 12 crayfish = 96 recordings.

**Figure 3 ijerph-17-02084-f003:**
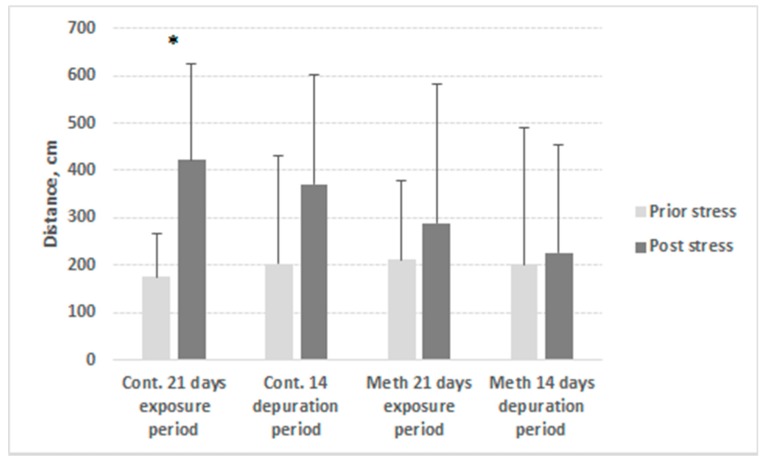
Mean distance moved by control (Ctr.) and METH-exposed (Meth) signal crayfish during exposure and depuration (D). Before (light grey) and after (dark grey) stress application. Columns correspond to the mean results of trials conducted on days 0, 1, 7, 14, and 21 of the 21-day exposure period and on days 1, 2, 7, and 14 of the 14-day depuration period. Whiskers correspond to standard deviation; n = 12 in each trial. * indicates significant difference (*p* < 0.05).

**Figure 4 ijerph-17-02084-f004:**
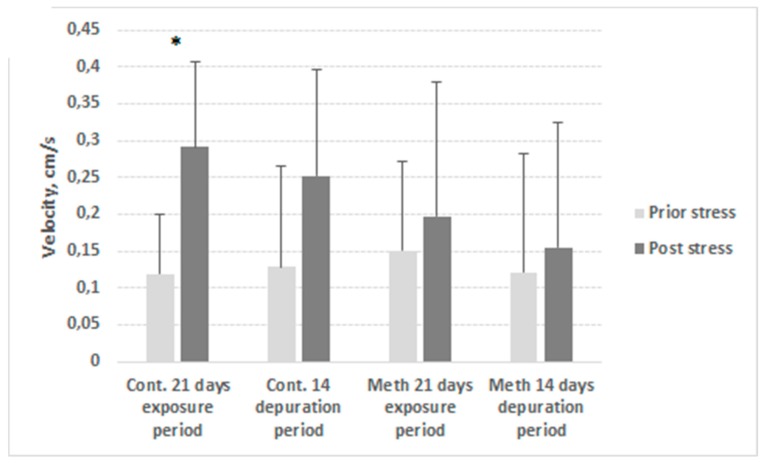
Mean velocity of control (Ctr.) and METH-exposed (Meth) signal crayfish during exposure and depuration (D). Before (light grey) and after (dark grey) stress application. Columns correspond to the means of trials conducted on days 0, 1, 7, 14, and 21 of the 21-day exposure period (21 days) and on days 1, 2, 7, and 14 of the 14-day depuration period. Whiskers correspond to standard deviation; n = 12 in each trial. * indicates significant difference (*p* < 0.05).

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
