# Peer review of "Cardiac and Locomotor Responses to Acute Stress in Signal Crayfish Pacifastacus leniusculus Exposed to Methamphetamine at an Environmentally Relevant Concentration"

_ijerph, 2020, doi:10.3390/ijerph17062084_

Round 1
Reviewer 1 Report
Good work, interesting and well managed.
Excellent presentation and nice discussion and conclusion.
Just one little thing I'd like to know (and perhaps to ad to the papaer) : what kind of food given, any measure to see if there is no contamination of it by similar pharmacocinetic molecule, what about the wastes (faeces)
Author Response
Good work, interesting and well managed.
Excellent presentation and nice discussion and conclusion.
Just one little thing I'd like to know (and perhaps to ad to the paper): what kind of food given, any measure to see if there is no contamination of it by similar pharmacocinetic molecule, what about the wastes (faeces)
Answer:
Dear reviewer, thank you for you kind evaluation and comments.
To your question, crayfish were fed on frozen chironomid larvae – added in to paper (line: 69). Uneaten food and waste were removed during water exchange the following day.
Place of chironomid larvae origin is not known, but usually it is periodic wetland without possible input of drugs residues. Anyway both groups were fed by larvae from the same source.
One way or another, water from control and exposure group was controlled for presence of contaminants. Control water was always under detection limits. So if food organisms were somehow contaminated it was not reflected in water samples and was very diluted. We expect that if the food is contaminated it is visible also in pollutant concentration in water – larvae were frozen and thawed i.e. their body fluids were free in the water.
Reviewer 2 Report
Reviewer's report for
"Cardiac and locomotor responses to acute stress in signal crayfish Pacifastacus leniusculus exposed to methamphetamine at an environmentally relevant concentration" (Manuscript ID: ijerph-731471) by Filip LoĹľek *, Iryna Kuklina, Katerina Grabicova, Jan Kubec, Miloš Buric, Tomáš Randák, Petr Císar, Pavel Kozák.
In this MS, the authors design and implement an experiment to analyze the cardiac and locomotor responses to acute stress in signal crayfish P. leniusculus during and following exposure to environmentally relevant
concentrations of METH and potential impacts of drug residues on a wider aquatic ecosystem, particularly with respect to predator-prey relationships. They found a statistically significant increase in HR for the METH-exposed group during the exposure time after stressor application (increase in 61% of recordings) versus 73% in the control group. They argue that that the increase in HR is significantly higher in the control group compared to the METH-exposed group. With regard to locomotor activity, the control group showed significant increase in distance moved and velocity in response to stress in the METH-free group, while the METH-exposed group showed no significant change in locomotor activity.
GENERAL APPRAISAL
The paper is relatively well written, but the statistical design and the statistical tests are not clearly described.
MAJOR POINTS
1. Lines 15 and 42: What does "non-target organism" stand for?
2. Line 60: How were the male and female Crayfish assigned to the treatment and the control group?
3. The sample size is too small for the statistical results to be conclusive.
4. Lines 116-120: What is the difference between " after stressor application" and "following stressor initiation"? The results as presented in this paragraph are confusing.
5. The authors report p-value without specifying the statistical test that were used.
6. Is 61% statistically different from 73% with such a small sample?
7. Line 127-129: Why was the single mouled Crayfish in the control group not included in the analysis?
8. Section 3.3: Please include in the x-labels of Figures 3 and 4 the "21-day Exposure period" and "14-day Depuration period".
MINOR POINTS
1. Line 30: … a wide variety of pharmaceutically...
2. Line 50: … promising model organism in investigating.....
3. Line 71:... to simulate predation, consistent with....
4. Line 163: ... from that of the controls, which....
5. Line 196: … could influence ....
Author Response
Reviewer's report for
"Cardiac and locomotor responses to acute stress in signal crayfish Pacifastacus leniusculus exposed to methamphetamine at an environmentally relevant concentration" (Manuscript ID: ijerph-731471) by Filip LoĹľek *, Iryna Kuklina, Katerina Grabicova, Jan Kubec, Miloš Buric, Tomáš Randák, Petr Císar, Pavel Kozák.
In this MS, the authors design and implement an experiment to analyze the cardiac and locomotor responses to acute stress in signal crayfish P. leniusculus during and following exposure to environmentally relevant concentrations of METH and potential impacts of drug residues on a wider aquatic ecosystem, particularly with respect to predator-prey relationships. They found a statistically significant increase in HR for the METH-exposed group during the exposure time after stressor application (increase in 61% of recordings) versus 73% in the control group. They argue that that the increase in HR is significantly higher in the control group compared to the METH-exposed group. With regard to locomotor activity, the control group showed significant increase in distance moved and velocity in response to stress in the METH-free group, while the METH exposed group showed no significant change in locomotor activity.
GENERAL APPRAISAL
The paper is relatively well written, but the statistical design and the statistical tests are not clearly described.
Response: Dear reviewer, thank you for you valuable comments and suggestions. We tried to fulfil all you request and recommendations to improve the scientific quality of the manuscript. Answers to your concrete comments are appended below.
MAJOR POINTS
- Lines 15 and 42: What does "non-target organism" stand for?
Response 1: Target organism of pharmaceuticals or synthetic drugs are humans or pets (human and veterinary pharmacology) usually mammals in general, any other like aquatic organism which are in contact with those compounds are “non-target” because they are treated unintentionally.
- Line 60: How were the male and female Crayfish assigned to the treatment and the control group?
Response 2: 50:50 sex ratio was used in both groups (added in to text – Line: 63)
- The sample size is too small for the statistical results to be conclusive.
Response 3: We used 12 individuals in both groups, that is relatively small “n”, but the nature of the research do not allow to use larger numbers. Firstly, the experimental work lasted several months including acclimation, exposure and depuration periods. During this time we analysed data from multiple samplings of heart rate and locomotor activity of exposure and depuration days. If we would repeat such long process again to increase “n” we probably can obtain different results because of time effect – crayfish during the year undergo different developmental stages as their moulting events or reproduction. Therefore is better to do smaller sample but compare the control and treatment at the same time.
- Lines 116-120: What is the difference between "after stressor application" and "following stressor initiation"? The results as presented in this paragraph are confusing.
Response 4: Thank you for your comment. It is only different expression for the same thing in methodology i.e. 30 mins after stress stimulus addition. We unified it in the text manuscript.
- The authors report p-value without specifying the statistical test that were used.
Response 5: I am not sure where exactly, statistical tests are mentioned in chapter materials and methods in section 2.2. Statistical analyses, where is described which tests were used for which result. If needed we can add concrete information to each p-value presented (e.g. as here: Kruskal-Wallis test, H = 128.5, p < 10-5) but we think it is redundant.
- Is 61% statistically different from 73% with such a small sample?
Response 6: The sample size is 48 measurements (12 individuals x 4 sampling days) for each period (exposure, depuration) per group. Please see supplementary materials. It is not written in results that these percentages significantly differ – it is only description of the individual crayfish results obtained, please see SM, Table 1, 2.
- Line 127-129: Why was the single mouled Crayfish in the control group not included in the analysis?
Response 7: Moulting process in crayfish (generally in arthropods) is very complicated and crayfish completely changing their behavioural and physiological patterns before, during and after it. We therefore consider this one moulted crayfish during experiments as not valid to be included to analyses in case of physiological changes leading to change of behaviour compared to crayfish in intermoult phase.
- Section 3.3: Please include in the x-labels of Figures 3 and 4 the "21-day Exposure period" and "14-day Depuration period”.
Response 8: added to figures
MINOR POINTS
Line 30: … a wide variety of pharmaceutically…
Response: changed in text
- Line 50: … promising model organism in investigating…..
Response: changed in text
- Line 71:... to simulate predation, consistent with….
Response: consisted in the meaning of scent consists from.. left unchanged
- Line 163: ... from that of the controls, which….
Response: changed in text
- Line 196: … could influence ….
Response: changed in text
Reviewer 3 Report
2.1. Experimental animals, experimental setup, data acquisition, chemicals: Please specify whether there was any shelter offered in the experimental tank.
Although the brook where the crayfish were collected from has no known source of possible drug residue input, it would make the study more sophisticated if the author also included concentrations of METH in water samples from the brook.
Please describe in detail whether the crayfish were in consistent conditions during treatments on different days. E.g. were the acute stress signals released, and parameters (heart rate and locomotion) recorded during the same period of the dark-light cycle? And were the crayfish fed/unfed during the treatments?
Reference:
L249: “Anguilla anguilla” should be italic
L255: “Dreissena polymorpha” should be italic
L258: “Dreissena polymorpha” should be italic
L277: “Anguilla anguilla” should be italic
L299: “Orconectes rusticus” should be italic
In this study the authors tested responses in heart rate, travel distance, and velocity of crayfish to acute stress during METH exposure. It’s an interesting and meaningful study and the manuscript is overall well written.
My major concern is that whether there’s clear evidence which could well justify using hemolymph of conspecific as an acute stress. In this study, the author used the scent of an injured conspecific to simulate predation, and further discussed the possible impact of METH on predator-prey relationship. On the other hand, crayfish is well known for their cannibalism behavior, the possibility of hemolymph from conspecific could serve as a feeding stimulus under certain conditions should not be ignored.
They way how hemolymph was prepared may also affect how crayfish respond in the test. In this study anaesthetised crayfish were used for preparing hemolymph solution; therefore, if the key alarm signals were mainly released by conscious crayfish, then the effect on tested crayfish here may not be able to reflect their response to predators.
There’s also difference among species. Hazlett (1994) found that the crayfish Orconectes propinquus did not show an alarm response to crushed conspecifics, while O. virilis could show strong feeding response, or a mixture of alarm and feeding responses to crushed crayfish, depended on how the crayfish was prepared.
Therefore, in my opinion the author would need more evidence, either from literature or observation, to confirm that hemolymph from anaesthetised signal crayfish used in this study can trigger similar response to predation in conspecifics. Otherwise, the conclusion about predator-prey relationship is not scientifically sound enough, and the manuscript may need some major revisions.
Author Response
Dear reviewer, thank you for you valuable comments and suggestions. We tried to fulfil all you requests and recommendations to improve the scientific quality of the manuscript. Answers to your concrete comments are appended below.
2.1. Experimental animals, experimental setup, data acquisition, chemicals: Please specify whether there was any shelter offered in the experimental tank.
Response: The tanks contained shelters made from half of a plastic plant pot with a narrow oblong opening cut in the upper part to ensure unimpeded entrance by the crayfish carrying sensors (added in to text – line: 65-67).
Although the brook where the crayfish were collected from has no known source of possible drug residue input, it would make the study more sophisticated if the author also included concentrations of METH in water samples from the brook.
Response: I understand your point. Crayfish were after collecting from brook transferred to recirculation system in our lab. Where they were for several weeks pre acclimatized in tap water where concentration of drugs is lower than the limit of quantification (< 0.04 μg L-1). Metamphetamine is in addition no longer in animal tissues than maximally weeks being secreted relatively fastly. Moreover, the brook is positioned outside the possible sources of pollution flowing from the National park Šumava in hills area. No contamination in control groups was detected by water samples analysis.
Please describe in detail whether the crayfish were in consistent conditions during treatments on different days. E.g. were the acute stress signals released, and parameters (heart rate and locomotion) recorded during the same period of the dark-light cycle? And were the crayfish fed/unfed during the treatments?
Response: Stressor was applied in the same time during light period of each sampling days. Crayfish were fed by chironomid larvae 24h before water changing (water was changed every 48h in both groups). Immediately before and during recording there was no feeding.
Reference:
L249: “Anguilla anguilla” should be italic
Response: changed in text
L255: “Dreissena polymorpha” should be italic
Response: changed in text
L258: “Dreissena polymorpha” should be italic
Response: changed in text
L277: “Anguilla anguilla” should be italic
Response: changed in text
L299: “Orconectes rusticus” should be italic
Response: changed in text
In this study the authors tested responses in heart rate, travel distance, and velocity of crayfish to acute stress during METH exposure. It’s an interesting and meaningful study and the manuscript is overall well written.
My major concern is that whether there’s clear evidence which could well justify using hemolymph of conspecific as an acute stress. In this study, the author used the scent of an injured conspecific to simulate predation, and further discussed the possible impact of METH on predator-prey relationship. On the other hand, crayfish is well known for their cannibalism behavior, the possibility of hemolymph from conspecific could serve as a feeding stimulus under certain conditions should not be ignored.
Response: Crayfish cannibalistic behaviour should not be overseen. However we used fluids of injured conspecifics to provide a scent associated with predation based on our previous experiences that has been shown it stimulates locomotion and cardiac activity in greater degree than other natural stressors i.e. food (Kuklina et al., 2018). Also from observation the reaction when crayfish were fed they were relatively excited, but in case of injured conspecific odour addition individuals showed avoiding/escsape reaction.
They way how hemolymph was prepared may also affect how crayfish respond in the test. In this study anaesthetised crayfish were used for preparing hemolymph solution; therefore, if the key alarm signals were mainly released by conscious crayfish, then the effect on tested crayfish here may not be able to reflect their response to predators.
Response: Small transparent plastic box contains crayfish was placed on ice. When the crayfish’s functions was slow down (10 min., not frozen) it was sacrificed by sagittal cut placed in to odour stock jar and all volume was gently shaken. Small amount of other body fluids i.e. urine which crayfish released in to small plastic box was washed also in to odour stock jar (we can add in to materials and methods section). In addition alarm responses of O. virilis was not eliminated even by adding by either freeze-thawing the conspecific (Hazlett, 1994).
There’s also difference among species. Hazlett (1994) found that the crayfish Orconectes propinquus did not show an alarm response to crushed conspecifics, while O. virilis could show strong feeding response, or a mixture of alarm and feeding responses to crushed crayfish, depended on how the crayfish was prepared.
Response: Hovewer O. propinquus show no alarm response to crushed conspecific (Hazlett, 1994), Kuklina et al. (2018) described higher cardiac excitation after addition odour of injured conspecific then odour of food. So the stressor effect was confirmed for signal crayfish as well as for method of preparation of the stressor cue. However we include this information to discussion section (line: 164-170) to make it clear for readers.
Therefore, in my opinion the author would need more evidence, either from literature or observation, to confirm that hemolymph from anaesthetised signal crayfish used in this study can trigger similar response to predation in conspecifics. Otherwise, the conclusion about predator-prey relationship is not scientifically sound enough, and the manuscript may need some major revisions.
Response: As written above, the effect of the stressor prepared by method described was confirmed by previous study. Hence, we can state that tested species is susceptible to the stressor used. To make it clearer we described it as it written above in our answer to your comment.
Reviewer 4 Report
In my opinion the topic is very interesting and up-to date. However, the manuscript suffers from major deficiencies, which I analyze below.My biggest concern is the description of experiment design. The authors took very small group of specimens for studies, so statistic test are proved unfounded. For such small sample size 12 females and males ("Twelve males and 12 females" in the text-please use numbers or description not both methods in one sentence) statistic test should be carefully chosen!What is more, the statement "The brook has no
known source of possible drug residue input" as we need to do chemical analysis to prove it!
Other suggestions:
Introduction: two general and not convincing e.g. "Drug residues may be integrated into tissues of aquatic invertebrates [16] and vertebrates [17]
and can modify the physiology of invertebrates [18-20]. They have been shown to alter behaviour of planarians [21,22], crayfish [23-27], and some aquatic vertebrates [28,29]". No details!
Results: No information about the test used, values of the test are not known, authors gave only p values e.g. "Significant increase (p < 0.05) of HR after stressor application was detected in 56% of recordings
in previously METH-exposed crayfish (Figure 2B) compared to 83% in the control group (Figure 2D) during depuration period." Significant increase is not sufficient!!
Conclusion
Please do not use citation in this part as it should be your thoughts.
Author Response
Dear reviewer, thank you for you valuable comments and suggestions. We tried to fulfil all you request and recommendations to improve the scientific quality of the manuscript. Answers to your concrete comments are appended below.
In my opinion the topic is very interesting and up-to date. However, the manuscript suffers from major deficiencies, which I analyze below.My biggest concern is the description of experiment design. The authors took very small group of specimens for studies, so statistic test are proved unfounded. For such small sample size 12 females and males ("Twelve males and 12 females" in the text-please use numbers or description not both methods in one sentence) statistic test should be carefully chosen! What is more, the statement "The brook has no known source of possible drug residue input" as we need to do chemical analysis to prove it!
Response: Dear reviewer, thank you for you valuable comments and suggestions. We tried to fulfil all you request and recommendations to improve the scientific quality of the manuscript.
To your questions, the number of crayfish is limited by our technical possibility however we are trying to extend classic ethological studies for physiological parameter.We used 12 individuals in both groups, that is relatively small “n”, but the nature of the research do not allow to use larger numbers. Firstly, the experimental work lasted several months including acclimation, exposure and depuration periods. During this time we analysed data from multiple samplings of heart rate and locomotor activity of exposure and depuration days. If we would repeat such long process again to increase “n” we probably can obtain different results because of time effect – crayfish during the year undergo different developmental stages as their moulting events or reproduction. Therefore is better to do smaller sample but compare the control and treatment at the same time.
Statistical tests are mentioned in chapter materials and methods in section 2.2. Statistical analyses, where is described which tests were used for which result. If needed we can add concrete information to each p-value presented (e.g. as here: Kruskal-Wallis test, H = 128.5, p < 10-5) but we think it is redundant.
Brook is situated in the hills of the natural park in addition crayfish were after collecting from brook transferred to recirculation system in our lab. Where they were for several weeks pre acclimatized in tap water where concentration of drugs is lower than the limit of quantification (< 0.04 μg L-1). Metamphetamine is in addition no longer in animal tissues than maximally weeks being secreted relatively fastly. Moreover, the brook is positioned outside the possible sources of pollution flowing from the National park Šumava in the hills area. No contamination in control groups was detected by water samples analysis.
Other suggestions:
Introduction: two general and not convincing e.g. "Drug residues may be integrated into tissues of aquatic invertebrates [16] and vertebrates [17] and can modify the physiology of invertebrates [18-20]. They have been shown to alter behaviour of planarians [21,22], crayfish [23-27], and some aquatic vertebrates [28,29]". No details!
Response: During writing of the manuscript was this part substantially shortened, because details are easily available in publications used, so the team of authors decided, that this information are more or less redundant. Below see the previous state of this part, if you insist on text extension we can offer this version:
“Drug residues may be integrated into tissue aquatic invertebrates and vertebrates as was described in crustaceans [16] and in fish [17]. Presence of drugs in aquatic environment can modify the physiology zebra mussels [18-20]. Ketamine and methaphetamine assess developmental toxicity, oxidative stress and behavioral alteration in early life stages as reported by Liao et al. (2015) [29] in fish Oryzias latipes during early development and in european eel Anguilla anguillla in which environmental concentration of cocaine caused endocrine disruption [28]. In addition, planarians [21,22] and crayfish [23-27] showed reinforcing effect to dopaminergic drugs similarly like target organism.
Results: No information about the test used, values of the test are not known, authors gave only p values e.g. "Significant increase (p < 0.05) of HR after stressor application was detected in 56% of recordings in previously METH-exposed crayfish (Figure 2B) compared to 83% in the control group (Figure 2D) during depuration period." Significant increase is not sufficient!!
Response: As mentioned above, it is almost the same comment. Statistical tests are mentioned in chapter materials and methods in section 2.2. Statistical analyses, where is described which tests were used for which result. If needed we can add concrete information to each p-value presented (e.g. as here: Kruskal-Wallis test, H = 128.5, p < 10-5) but we think it is redundant, because of description of tests used.
Conclusion
Please do not use citation in this part as it should be your thoughts.
Response: Citation removed
Round 2
Reviewer 2 Report
Reviewer's report for the revised version of
"Cardiac and locomotor responses to acute stress in signal crayfish Pacifastacus leniusculus exposed to methamphetamine at an environmentally relevant concentration" (Manuscript ID: ijerph-731471) by Filip LoĹľek *, Iryna Kuklina, Katerina Grabicova, Jan Kubec, Miloš Buric, Tomáš Randák, Petr Císar, Pavel Kozák.
GENERAL APPRAISAL
The authors have satisfactorily addressed my concerns with the previous version of the MS. I think that the current version of the MS is suitable for publication pending a minor revision. I would particularly urge the authors to check the English grammar and vocabulary.
SPECIFIC COMMENTS
1) Your response to question 1 of my previous reviewer’s report was along the lines :” Organisms of pharmaceuticals or synthetic drugs are humans or pets, usually mammals or any other organism like aquatic organisms which are in contact with those compounds are called non-target organisms because they are treated unintentionally.
Please include this statement after the first mention of the term non-target organism since this may not be clear to every reader.
2) With regard to your response to my question 3 of the previous reviewer’s report which reads “We used 12 individuals in both groups, that is relatively small “n”, but the nature of the research do not allow to use larger numbers. Firstly, the experimental work lasted several months including acclimation, exposure and depuration periods. During this time, we analyzed data from multiple samplings of heart rate and locomotor activity of exposure and depuration days. If we would repeat such long process again to increase “n” we probably can obtain different results because of time effect – crayfish during the year undergo different developmental stages as their moulting events or reproduction. Therefore is better to do smaller sample but compare the control and treatment at the same time.
A question that spring to mind is, can the time effect be controlled for, for instance through the inclusion of relevant environmental covariates?
Author Response
GENERAL APPRAISAL
The authors have satisfactorily addressed my concerns with the previous version of the MS. I think that the current version of the MS is suitable for publication pending a minor revision. I would particularly urge the authors to check the English grammar and vocabulary.
Answer: Dear reviewer thank you for your statement about revised manuscript. I would like to kindly mention that manuscript was under language correction by native speakers (Lucidus Consultancy) but we will check it again.
SPECIFIC COMMENTS
1) Your response to question 1 of my previous reviewer’s report was along the lines :” Organisms of pharmaceuticals or synthetic drugs are humans or pets, usually mammals or any other organism like aquatic organisms which are in contact with those compounds are called non-target organisms because they are treated unintentionally.
Please include this statement after the first mention of the term non-target organism since this may not be clear to every reader.
Answer: Added in introduction section – line: 42.
2) With regard to your response to my question 3 of the previous reviewer’s report which reads “We used 12 individuals in both groups, that is relatively small “n”, but the nature of the research do not allow to use larger numbers. Firstly, the experimental work lasted several months including acclimation, exposure and depuration periods. During this time, we analyzed data from multiple samplings of heart rate and locomotor activity of exposure and depuration days. If we would repeat such long process again to increase “n” we probably can obtain different results because of time effect – crayfish during the year undergo different developmental stages as their moulting events or reproduction. Therefore is better to do smaller sample but compare the control and treatment at the same time.
A question that spring to mind is, can the time effect be controlled for, for instance through the inclusion of relevant environmental covariates?
Answer: Dealing with problem theoretically sounds really nice, unfortunately practically, in case of working with wild caught animals in laboratory controlled conditions, there will be always conflict between their natural rhythm (seasonal conditions) and stable experimental conditions, with unpredicted consequences to their physiology and behaviour. From its nature it is probably more preferable to use in experimental design only with control vs exposure group simultaneously in the same time.
Reviewer 3 Report
Thanks for the author's detailed responses to my questions and concerns. I think my questions and thoughts have been explained well by the author, and necessary revisions and information were added; the manuscript is definitely in better shape now. I would feel comfortable to accept the manuscript for publication.
Author Response
Thanks for the author's detailed responses to my questions and concerns. I think my questions and thoughts have been explained well by the author, and necessary revisions and information were added; the manuscript is definitely in better shape now. I would feel comfortable to accept the manuscript for publication.
Answer: Dear reviewer thank you for your statement based on our previous communication.
Reviewer 4 Report
The authors replied for the remarks but they did not ameliorate the manuscript significantly. I think that the information about test and p values are basic information and should be into the text. The same remark refers to the description of the brook, it should be placed in the methodology. The proposition of previous version of introduction is a good idea also.
Author Response
The authors replied for the remarks but they did not ameliorate the manuscript significantly. I think that the information about test and p values are basic information and should be into the text. The same remark refers to the description of the brook, it should be placed in the methodology. The proposition of previous version of introduction is a good idea also.
Answer: Thank you for reviewer´s noticing, the informations about tests are written in materials and methods. We would like to put p values in to text as reviewer suggested, unfortunately we are not sure where (in our opinion p values are already in the text), if there are uncertainties please remark specific place in text so we can put required informations.
Information about brook were added in methodology chapter - line 63-64.
Proposition from previous version of manuscript was added in to introduction chapter – line 43-48.